# Objective Patterns of Face Recognition Deficits in 165 Adults with Self-Reported Developmental Prosopagnosia

**DOI:** 10.3390/brainsci9060133

**Published:** 2019-06-06

**Authors:** Sarah Bate, Rachel J. Bennetts, Nicola Gregory, Jeremy J. Tree, Ebony Murray, Amanda Adams, Anna K. Bobak, Tegan Penton, Tao Yang, Michael J. Banissy

**Affiliations:** 1Department of Psychology, Bournemouth University, Poole BH12 5BB, UK; ngregory@bournemouth.ac.uk (N.G.); emurray@bournemouth.ac.uk (E.M.); adamsa@bournemouth.ac.uk (A.A.); a.k.bobak@stir.ac.uk (A.K.B.); 2College of Health and Life Sciences, Division of Psychology, Brunel University, Uxbridge UB8 3PH, UK; Rachel.Bennetts@brunel.ac.uk; 3Department of Psychology, Swansea University, Swansea SA2 8PP, UK; J.Tree@swansea.ac.uk; 4MRC Social, Genetic and Developmental Psychiatry Centre, Institute of Psychiatry, Psychology and Neuroscience, King’s College London, London SE5 8AF, UK; tegan.penton@kcl.ac.uk; 5Department of Psychology, Tsinghua University, Beijing, 100084, China; t_yang@mail.tsinghua.edu.cn; 6Department of Psychology, Goldsmiths, University of London, London SE14 6NW, UK; m.banissy@gold.ac.uk

**Keywords:** prosopagnosia, face recognition, face perception, individual differences

## Abstract

In the last 15 years, increasing numbers of individuals have self-referred to research laboratories in the belief that they experience severe everyday difficulties with face recognition. The condition “developmental prosopagnosia” (DP) is typically diagnosed when impairment is identified on at least two objective face-processing tests, usually involving assessments of face perception, unfamiliar face memory, and famous face recognition. While existing evidence suggests that some individuals may have a mnemonic form of prosopagnosia, it is also possible that other subtypes exist. The current study assessed 165 adults who believe they experience DP, and 38% of the sample were impaired on at least two of the tests outlined above. While statistical dissociations between face perception and face memory were only observed in four cases, a further 25% of the sample displayed dissociations between impaired famous face recognition and intact short-term unfamiliar face memory and face perception. We discuss whether this pattern of findings reflects (a) limitations within dominant diagnostic tests and protocols, (b) a less severe form of DP, or (c) a currently unrecognized but prevalent form of the condition that affects long-term face memory, familiar face recognition or semantic processing.

## 1. Introduction

Developmental prosopagnosia (DP) is a neurodevelopmental condition characterised by a severe deficit in the recognition of facial identity, which is thought to result from a failure to develop the necessary visuo-cognitive mechanisms [1]. Estimates suggest that DP affects approximately 2% of the adult population [2] and 1–4% of children [3]. While these figures are a product of the statistical techniques used to infer impairment [4,5], they also result from the diagnostic protocols adopted in each study. The latter reflects the general state-of-the-art, whereby vast differences in inclusion criteria prohibit many cross-study comparisons. As a result, it is often unclear whether differences in cognitive presentation result from inconsistencies in neuropsychological screening or theoretically important heterogeneity between individuals.

The first case report of DP was presented by McConachie in 1976 [6], and over 150 papers have subsequently been published. Increased interest in DP coincided with the development of the Internet and large-scale media coverage of the condition, allowing large numbers of people to contact researchers in the self-belief that they experience severe face recognition difficulties. A wide variety of techniques have been used to assess these individuals (for a recent review, see [7]), with a minority of authors relying on self-reported anecdotal evidence alone (e.g., [8]). While respectable associations have more recently been reported between multi-item subjective ratings of face recognition ability and objective measures (e.g., [9]), it is nevertheless clear that some “typical” performers erroneously under-rate their face recognition ability and self-refer for DP assessment [10,11]. Most authors therefore agree that self-rating or anecdotal evidence of DP should always be supplemented by objective measures of face recognition performance [4,12], although the proportion of confirmed diagnoses versus false alarms has not yet been examined in a large group of individuals.

Using objective measures to identify cases of DP has followed the precedent set by the longstanding literature on acquired prosopagnosia. Traditionally, the acquired condition was confirmed using famous face tests (e.g., [13,14]) or bespoke assessments composed of personally familiar faces (e.g., [15]). These tests were later supplemented with standardized tests of unfamiliar face memory (e.g., using the Recognition Memory Test for Faces: [16]) or perception (e.g., using the Benton Facial Recognition Test: [17]). While famous face tests remain popular for the assessment of DP (e.g., [18,19]), bespoke tests of personally familiar face recognition are labour-intensive to prepare for the typically larger samples of developmental compared to acquired cases, and it is also difficult to collect well-matched control data. Further, both the Warrington and Benton tests have been subject to some criticism, as successful recognition performance can be achieved using extra-facial cues alone (e.g., the hairline or clothing: [20]). These tests have therefore been superseded by the Cambridge Face Memory Test (CFMT: [21]) and Cambridge Face Perception Test (CFPT: [22]). While the CFMT assesses participants’ ability to encode six previously unknown faces and to subsequently recognize these individuals from memory, the CFPT has no mnemonic demands, requiring participants to sort similar images of simultaneously-presented faces in relation to a target.

Together, tests of face perception, face memory and famous face recognition offer a theoretically-driven assessment battery, and acquired deficits have correspondingly been reported (and occasionally dissociated) at perceptual (e.g., [14,23]), mnemonic (e.g., [14,24]) and semantic (e.g., [25]) stages of processing. Evidence that face perception can sometimes be preserved in acquired prosopagnosia has had particular theoretical impact, suggesting its independence from face memory. As such, this process is represented at an earlier, distinct stage in dominant theoretical models of face-processing (e.g., [26,27]), and apperceptive and associative subtypes of acquired prosopagnosia have been proposed [28]. 

Given that developmental disorders often present with heterogeneous profiles of cognitive and perceptual deficits (e.g., [29,30,31]), and evidence supporting the independence of face perception and face memory has also been found in typical children [3,32], it seems likely that cases of DP are similarly variable in their presentation. That is, we would not expect all individuals with DP to present with a significant perceptual impairment. However, evidence from developmental cases is sparse and contradictory, and it remains unclear whether the two processes unfold independently (for an overview, see [33]). Although some adult DP case studies support the independence of face perception and memory (e.g., [34]), larger case series disagree about the prevalence of dissociable deficits. For instance, Dalrymple and colleagues [35] used conservative single-case statistics to report dissociations between intact CFPT and impaired CFMT performance in 6 out of 16 DPs, whereas a very recent study used group-based analyses on the same measures to suggest a much greater overlap in impairment in a larger sample of participants [36]. Without additional data, it is difficult to reconcile the findings of these two studies given the differences in sample size and statistical approach. 

Further, almost no work to date has considered whether isolated semantic-level impairments can present in DP, as has been reported for acquired cases. Bowles et al. [2] raise this possibility in an individual who self-reported severe everyday face recognition difficulties and was impaired at famous face recognition, but achieved typical scores on the CFMT and CFPT. While it is possible that this person used compensatory strategies to obscure impairment on the latter two tests (see [21]), evidence from another case study suggests that long-term memory for unfamiliar faces can be independently impaired in development. McKone et al. [37] reported an individual who also achieved typical CFPT and CFMT-Aus (an alternate version of the standard CFMT) scores in the context of impaired famous face recognition, and additionally performed poorly on long-term recall (at 20-min and 24-h delays) of the CFMT-Aus faces. As this pattern of presentation has not yet been systematically examined, it remains unclear whether further, as yet unidentified, subtypes of DP may exist. 

Understanding the different patterns of presentation that can exist within the DP umbrella has important implications for current screening protocols: many authors administer the CFPT, CFMT and a famous faces test at screening, and adhere to the recommendation that impairment should be noted on at least two tests for DP diagnosis [4,12]. This recommendation for repeated testing is important to minimize false alarms, reducing “the chance that it happened by chance” ([14], p. 945). Yet, the above review indicates that some individuals may genuinely only be selectively impaired on one test, and would not meet current criteria for DP—prohibiting both theoretical and clinical developments in our understanding and management of the condition.

The current study aimed to examine the substantive issues associated with the diagnosis of DP, as outlined above. We screened the largest sample of self-reported DP cases that has been reported to date, using the dominant battery of assessment tests that is currently used by most researchers in the field (i.e., the CFMT, CFPT and a famous faces test). First, we examined the proportion of cases that meet existing diagnostic criteria for DP (i.e., impairment on any two tests). Second, we examined whether specific patterns of dissociation that have been observed in acquired cases (i.e., preserved versus impaired face perception, and selectively-impaired famous face memory) occur in DP participants. Importantly, we also note the patterns of presentation that *do not* occur, and their implications for theoretical models of face recognition. As a supplement to these analyses, we also examine the utility of each diagnostic test across our target age range (18–78 years), and present UK age-appropriate norms that will be of widespread value to the field.

## 2. Methods

### 2.1. Participants

Data were collected from 195 adults (61 male, age range 18–77, M = 48.2, SD = 14.2) who self-referred to our laboratory in the belief that they have DP. All participants were asked, prior to the testing session, to declare any history of neurological, psychiatric or socio-emotional disorder. Only participants who did not declare any such condition were invited to participate in the study. A total of 241 control participants (100 male, age range 18–78, M = 49.7, SD = 18.3) were also tested, and subjected to the same exclusion criteria. As per the recommendations of Bowles et al. [2] they were subdivided into five separate groups (see Table 1) to provide age-appropriate data for individual comparisons. Control participants (but not DPs) were offered a small financial incentive in exchange for their time. Ethical approval for the study was granted by the institutional Ethics Committee, informed consent was provided by all participants prior to data collection, and the study was carried out according to the rules of the 1975 Declaration of Helsinki.

### 2.2. Materials

#### 2.2.1. Background Neuropsychological Screening

*IQ*: The IQ of DP and control participants was estimated using the Wechsler Test of Adult Reading (WTAR; [38]). A cut-off of 70 [38] was used to directly exclude any individuals with low IQ. Given IQ is not associated with face-processing skills [39,40], a more conservative cut-off was not deemed to be necessary. To detect atypical cognitive decline in older participants, controls and DPs aged more than 65 years also completed the Mini-Mental State Examination (MMSE: [41]). The recommended cut-off score of 26 [42] was used to determine any exclusions.

*Lower-level vision*: Lower-level vision was assessed in the self-referred DP group (but not controls) in order to check whether the participants’ difficulties in face recognition were underpinned by basic perceptual impairments. Four sub-tests from the Birmingham Object Recognition Battery (BORB: [43]) that have been used in previous investigations (e.g., [44]) were selected. In the Length Match test, participants are required to judge whether two lines are of the same length; in the Size Match test, they judge whether two circles are of the same size; in the Orientation Match test, they decide whether two lines are parallel or not; and in the Position of the Gap Match test, they decide whether the position of the gap in two circles is in the same place or not. Published cut-offs [43] were applied.

*Basic category recognition*: Barton and Corrow [4] suggest that general visual agnosia can be excluded by assessing object recognition at a basic category level, whereas debates about DPs’ abilities to distinguish between specific items within the same category are ongoing and should not be exclusion criteria. General visual agnosia was therefore excluded in the DP participants via the Object Decision Test (hard version) of the BORB, using published cut-offs [43]. We also enquired (but did not formally assess) whether DPs also experienced problems with voices or names, to exclude a multi-modal problem with person recognition (e.g., [45]). 

*Concurrent socio-emotional disorder*: Given face recognition difficulties can also present in autism spectrum disorder (ASD; [46]), we screened all participants using the Autism Quotient (AQ; [47]), a 50-item self-report questionnaire. The AQ is not a formal diagnostic tool for identifying ASD, but very few age-matched controls score higher than 32 on the questionnaire [47]. As such, we excluded data from any participant scoring higher than 32 from the current analysis, to minimise the possibility that participants presented with a socio-emotional disorder in addition to their face recognition problems. 

#### 2.2.2. Face-Processing

*Face perception:* The CFPT [22] is a standardised test that is used to measure face perception skills. In each trial, participants are asked to order a set of six comparison faces based on their similarity to a single test face, where each of the comparison faces has been morphed towards the test face by different degrees. While the CFPT contains eight upright and eight inverted trials, we followed the precedent of many previous investigations (e.g., [48,49]) by only examining the upright items (inverted items were not presented). For each trial, the final sorted order is scored by summing the deviations from the correct order (e.g., if a face is five places away from its proper place, it contributes 5 to the score). A score of 0 represents perfect performance, while the maximum possible score is 144. To aid in the analysis, these scores were converted into percentage correct using the formula [100 × (1 − (score/maximum score))] [50].

*Unfamiliar face recognition:* The CFMT is a measure of unfamiliar face memory that demonstrates high reliability and both convergent and divergent validity [2,51]. In the first part of the task, participants are introduced to six target faces and are then tested with forced-choice items consisting of three faces, one of which is a target. For each target face, three test items contain views identical to those studied in the introduction, five present novel views, and four present novel views with noise. At two points in the test, participants are given the opportunity to review the target faces before proceeding with the next set of trials (for full details see [21]).

*Familiar face recognition:* We used two different versions of a famous faces test (see our previous work: e.g., [44]), one for adults aged 35 years and older, and another for younger adults (those aged 18–34 years). Each version contains the faces of 60 celebrities, selected via initial pilot-testing where we ensured the familiarity of the faces for our target age groups. Faces were displayed sequentially, in a random order, for an unlimited time period. A correct identification was scored by provision of the celebrity’s name, or uniquely identifying biographical information about that person. If a participant could not identify a face, they were subsequently told who that person was, and asked if they have had previous exposure to that individual. Any celebrities that were unknown to each participant by name or biographical information were removed from the overall score and percentage correct was adjusted accordingly.

### 2.3. Procedure

DP and control participants completed the battery of tests in face-to-face laboratory settings. Testing sessions began with an initial semi-structured interview. During this process, control participants provided basic demographic information, and were also asked if they had any history of neurological, psychiatric or developmental disorder, or had experienced any periods of visual abnormality. The same questions were asked of DP participants, who were also invited to share anecdotal evidence of their face recognition difficulties. Participants then completed the relevant generalised and face-processing tests, in the order that they are described above.

### 2.4. Statistical Analyses

After initial exclusions were applied to the DP group (according to medical history, IQ, MMSE, BORB and AQ scores), the face-processing performance of each remaining individual was assessed relative to age-matched controls. For each test, impaired performance was deemed to be an accuracy score that significantly differed from control performance on modified *t*-tests for single-case comparisons ([52]; see Appendix A). Patterns of impairment were then considered against existing diagnostic criteria [4,12].

We then examined whether (a) face perception can be preserved relative to both short- and long-term face memory impairments (i.e., comparing CFPT scores to CFMT and famous face performance), and (b) long-term face memory can be selectively impaired (i.e., comparing famous face to CFMT and CFPT scores). We first examined these issues at the group level, via a principal components analysis on the entire DP sample (i.e., regardless of whether each individual met any particular inclusion criteria), and via correlational data. To assess whether more formal dissociations exist in individual DPs, Crawford and Garthwaite’s [53] Bayesian Standardized Difference Test (BSDT) was used to estimate whether each individual’s standardized difference between the specified test scores differed from that observed in the relevant control sample. Where required, the sequential Bonferroni procedure was applied to correct for multiple comparisons. 

## 3. Results

### 3.1. Control Performance

No exclusions were made to the control group according to neuropsychological history, IQ, MMSE or AQ scores (most had been pre-screened according to neuropsychological background prior to recruitment for the current study). Because all controls self-reported normal face recognition skills in everyday life, we retained all their data on the face-processing tests as we were interested to see whether any individual would inadvertently meet the criteria for DP. This occurred in one individual (a 74-year-old female) who was impaired on both the CFPT and CFMT (see Appendix A). Seventeen other individuals were impaired on any one test (see Table 2 for summary data, and Appendix A for individual test scores and *p* values), but no control participant was significantly impaired on all three tests. It should be noted that, as observed in previous work (e.g., [2]), control norms on all tests decrease according to age, with particularly large decreases on all tests for the 70–79 age-group (see Table 2). Statistically, such low means and large SDs mean that very low test scores are required for detection of significant impairment.

### 3.2. DPs: Initial Exclusions

Upon further interview at the testing session, two individuals described a severe psychiatric history (23- and 42-year-old females), and one 72-year-old male reported neurological injury. Twenty-nine further participants (aged 19–69, M = 48.3, SD = 13.4) achieved high scores (>32) on the AQ, and were also excluded from the study. Twenty-five participants chose to not complete the AQ, but their data were retained given no individual self-described autism traits. No exclusions were made on the basis of low-level visual assessment, IQ (three DPs were not tested due to dyslexia: two in the 35–49 age-group and one in the 50–59 age-group), MMSE (five declined this test but achieved typical scores on all other background screening tests), or BORB scores (eight participants did not complete these tests but were able to successfully complete the other tasks). No DP reported co-morbid difficulties with voices or names. After all exclusions, the final sample size of the DP group was 165 (111 female) individuals aged between 18 and 77 years (M = 48.2 years, SD = 14.2; see Table 3). No significant differences were observed between control and DPs according to IQ in any age group (all *p* > 0.05).

### 3.3. Diagnosis of DP

A total of 33 participants (20.0%) did not score in the impaired range on any test, fairly equally distributed across the five age groups (see Table 3). One individual (from the 50–59 age group) scored in the impaired range on the CFPT alone, and five participants were only impaired on the CFMT (one in the 18–34 age group, and four in the 50–59 age group). Sixty-three participants, representing all age-groups, were only impaired on the famous faces test (see Table 3).

The remaining 63 participants (38.2% of the final sample) met the widely accepted criteria for DP: impaired performance on at least two of the three tests [12]. Twenty participants performed poorly on all tests, predominantly (*N* = 11) from the 35–49 age group (see Table 3). Forty-three further participants were impaired on any two tests: two on the CFPT and CFMT, 10 on the CFPT and famous faces test, and 31 on the CFMT and famous faces test (see Table 3 for age distributions).

### 3.4. Dissociating Different Patterns of DP Presentation

Data from the entire self-referred DP group (*N* = 165) were entered into a principal components analysis (PCA) to explore overall patterns of association between the CFPT, CFMT and famous faces test. Solutions for one and two factors were each examined using varimax and oblimin rotations of the factor loading matrix. The two factor varimax solution (which explained a total of 83.52% of the variance) was preferred as it offered the best defined factor structure (see Table 4). The first factor had high loadings from the CFPT and CFMT, whereas the second factor only had a high loading from the famous faces test. A full correlation matrix is displayed in Table 5.

To investigate the relationship between face perception and face memory, we initially considered associations at the group level. CFPT performance correlated with both CFMT and famous face scores (see Table 5), and loaded highly on the same factor as the CFMT in our PCA (see Table 4 and Figure 1). While these findings indicate some overlap in the processes tapped by the tests at the group level, particularly between the CFPT and CFMT, we nevertheless proceeded to more specifically examine whether face perception can be intact in some individual cases. As such, BSDT analyses were carried out on the 31 cases with impaired CFMT and famous face scores but intact CFPT performance. Significant dissociations were observed for both face memory tests in four individuals (see Table 6).

Next, we examined whether long-term face memory (i.e., famous face performance) can be selectively impaired. At the group level, the PCA indicated that famous face recognition loaded on a separate factor to CFPT and CFMT scores (see Table 4), although mild significant correlations were observed with both tests (see Table 5 and Figure 2). At the single case level, BSDT analyses were carried out on the 63 participants who were only impaired at famous faces, comparing these scores to both CFPT and CFMT performance. Forty-two of these individuals showed significant dissociations with both tests (see Appendix A).

In order to examine the relationship between famous face, CFMT and CFPT performance in this group of individuals, we initially refined the group to those participants aged below 60 years (i.e., before scores more significantly decline according to age, particularly on the CFMT, and hence where interpretation is more reliable). In the 36 remaining individuals, famous face scores did not correlate with either CFPT (*r* = 0.16, *p* = 0.0353) or CFMT (*r* = 0.28, *p* = 0.105) performance (see Figure 3), nor was there a significant correlation between the latter two tests (*r* = 0.18, *p* = 0.291). As Figure 3 indicates, while many DPs performed below the control mean on the CFMT, just under half of the participants performed within one SD. However, a third of the participants achieved positive *z* scores on the CFPT, and the majority of the remaining participants performed within one SD of the control mean.

## 4. Discussion

This investigation examined patterns of face recognition impairments in 165 adults who self-referred for DP. A total of 38.2% of the sample met standard diagnostic criteria of impairment on at least two out of three dominant tests. Conservative single-case analyses found dissociations between face perception and both unfamiliar and familiar face recognition tests in four individuals who met DP criteria (6.3% of the overall sample who met DP criteria). More surprisingly, a dissociation between impaired famous face recognition and both unfamiliar face memory and face perception was observed in a large proportion (25.5%) of the sample. While these individuals do not reach existing diagnostic criteria for DP, a number of possible interpretations of this finding have implications for developmental theories of face recognition, our understanding of DP, and existing diagnostic protocols.

First, we provide evidence to support the independence of face perception and face memory in DP, using conservative criteria that required face perception to be intact relative to impaired performance on both the CFMT and famous faces test. This finding supports parallel findings in acquired prosopagnosia (e.g., [24]; see also [54]) and previous DP case studies (e.g., [55]), suggesting that face perception and face memory follow at least partially independent developmental trajectories that can be selectively impaired [32]. Interestingly, the wider patterns of impairment reported here support linear accounts of face-processing (suggesting that early impairments in face perception likely result in downstream impairments in face memory): performance of the one individual who was only impaired at face perception did not dissociate from performance on either face memory test. Likewise, while there were two individuals who showed impairments on the CFMT and CFPT, but not the famous faces test, their performance with famous faces did not significantly dissociate from the CFMT or CFPT scores. In short, none of the 165 cases examined here show evidence of being able to compensate for abnormal perceptual processing or short-term face encoding, even with high levels of exposure to a face.

However, perhaps most interesting is the low prevalence of mnemonic cases of DP in our large sample (i.e., in contrast to the six out of 16 cases reported by Dalrymple et al.: [35]), as traditionally defined by intact CFPT performance but impaired CFMT and/or famous faces scores. This may result from our conservative comparison of CFPT to *both* CFMT and famous face performance, whereas Dalrymple and colleagues only compared to the CFMT. However, if we include the cases who showed a dissociation between intact CFPT performance and impaired CFMT scores (but not famous faces), only an additional four cases would have been included. On the one hand, the low prevalence of mnemonic cases reported here fits well with the recent work of Biotti and colleagues [36], who used group analyses to suggest a widespread overlap between impaired face perception and unfamiliar face memory in DP—a finding that is also supported by our PCA. Together, these findings suggest that the CFPT and CFMT tap similar processes, and that unfamiliar face memory impairments in DP are mostly (but not always) underpinned by at least some weakness in face perception (affecting the encoding of faces; see also [19]). However, it is likely that this widespread weakness in face perception may not exceed diagnostic cut-offs in many cases (31 of the 63 DPs who met existing diagnostic criteria in this study did not achieve impaired scores on the CFPT). While this finding may result from the poorer psychometric properties of the CFPT ([2,56]; also see discussion below about age effects), it is equally likely that impairments in face perception are a matter of degree, rather than absolute. Such a suggestion is not new, and was offered by Farah [57] in her account for apperceptive versus associative acquired visual agnosia. Indeed, Farah suggests that, when tested appropriately, most visual agnosia patients show perceptual deficits, but these reside on a continuum. It is therefore conceivable that the same pattern occurs in DP.

However, this explanation does not sit well with our finding of selectively impaired famous face recognition, compared to both CFPT and CFMT performance, in a large number of individuals. Indeed, if DP can be explained by a unitary hypothesis of suboptimal face perception skills, it seems unlikely that these deficits were not also detected by a test that is solely intended to tap face perception skills (the CFPT), nor a test of short-term unfamiliar face memory that contains an initial perceptual encoding phase (the CFMT). One could argue that successful performance can be achieved on both the CFPT and CFMT using suboptimal strategies (e.g., feature matching), and such compensatory mechanisms are less effective in a test of famous face recognition that uses single presentations of static images. However, this argument is countered by a lack of correlation between famous face and CFMT/CFPT scores in this group of individuals (see Figure 3), and evidence of the reverse dissociation (i.e., intact famous face recognition that is significantly dissociated from impairments scores on both the CFPT and CFMT) in one DP. That is, an individual who performed within the typical range on a test that is more difficult to “cheat”, but not on two tests that are known to be susceptible to compensatory strategies. In addition, existing ERP evidence indicates an intact N250 component (thought to reflect activation of stored memory traces of individual faces) in six DPs viewing famous faces, but an impaired P600f component, thought to be linked to the semantic stages of identification [58]. Likewise, previous work from our group has reported typical patterns of eye-movements in a DP with no known deficits in face perception when viewing famous faces [59].

Alternatively, our findings can be seen to provide additional weight to previous indications from single-case studies that long-term face memory [37], or memory for familiar faces [2], may be selectively impaired. Strikingly, evidence of this dissociation in terms of impaired famous face recognition was not only observed in isolated cases in the current study, but in rather a large proportion of our sample: while 63 of the 165 individuals were only impaired at famous face recognition, this dissociation reached significance with regard to both CFMT and CFPT performance in 42 cases (reduced to 36 when cases that could be influenced by age-related decline in control norms were removed). However, it cannot be concluded from the available data whether at least some of these individuals represent a subtype (or even multiple subtypes) of DP that is not yet recognised, and further exploration is required to interpret the precise functional and cognitive underpinnings of the deficit. Indeed, the impairment may be in creating detailed and stable perceptual representations of familiar individuals, in retaining these representations in long-term memory, or in storing, linking or accessing semantic information about familiar people. Of course, all these possibilities may independently exist, and the potential diagnostic entities that reside between the processes tapped by the CFMT and famous faces test are currently unknown. One possibility that can be excluded here is that any famous face recognition deficits resulted from a developmental form of proper name anomia, as we accepted biographical information as a correct response in the absence of naming. While no developmental case of proper name anomia has been reported to date, it is also pertinent that we did not detect any potential candidates for the condition in the large sample reported here.

Importantly then, the above discussion casts the existing array of diagnostic tools and protocols into question. While many would agree that the field lacks a reliable test of face perception (e.g., [60]), it is also generally well-accepted that intact scores can sometimes by achieved on the CFMT via compensatory mechanisms. Famous face tests are not without fault: they may be undermined by overly liberal cut-offs that result from ceiling effects and small SDs in controls, and can also be confounded by variations in lifetime exposure to target celebrities. In the current study, we attempted to address the latter by removing low familiarity trials from overall scores on an individual basis, and we used conservative case-by-case statistics to indicate both impairment and dissociation. Nevertheless, there were six false alarms in controls on the famous face test (compared to two on the CFMT and 11 on the CFPT), although this proportion of isolated famous face recognition impairments is clearly much lower than in the DP sample. One way of dealing with these issues is to allow more liberal cut-off scores for the CFMT and CFPT to capture cases of compensatory processing, while simultaneously making the famous face cut-off more conservative to ensure that “atypical” scores are truly impaired rather than suboptimal. It is likely that allowing more liberal cut-offs on the CFMT and CFPT would be controversial to the field, and may introduce a higher false-positive diagnosis rate. On the other hand, using more conservative cut-offs for the famous faces test may be more practical. Due to the format of the famous faces test, it is extremely unlikely that an individual can perform well by randomly guessing. Consequently, the effective “floor” of performance that can be meaningfully interpreted is much lower than in tasks which require forced responses, such as the CFMT and CFPT. Pertinently, Figure 3 indicates that even if we dropped the cut-off to seven SDs below the control mean, we would still find cases to support the impaired famous face dissociation compared to intact CFPT and CFMT performance. That is, some DPs scored −7 SDs below the control mean on the famous faces test, yet achieved positive *z* scores on both the CFPT and CFMT). Not only do these individuals support the novel dissociation evidenced here, but also demonstrate the large range of impaired performance that can be tapped by a famous faces test, and perhaps used to grade the severity of face recognition impairments in a way that the CFMT cannot.

It is also pertinent that famous face tests have traditionally been the preferred means of diagnosing acquired cases of prosopagnosia, as impairments in familiar face recognition tend to be more striking and robust than those of unfamiliar face recognition. Indeed, the high mean scores and small standard deviations of control participants reported here (even for older participants) make very low scores particularly salient. It is therefore possible that famous face tests currently offer a more accurate assessment for prosopagnosia than tests of unfamiliar face recognition, perhaps due to (a) their greater ecological validity, (b) because familiar face recognition is less affected by vast individual differences in the typical population, (c) because familiar faces are less affected by extraneous cues and non-identity-related changes to images (e.g., viewpoint or lighting; [61]) than unfamiliar face recognition, (d) they largely prohibit compensatory processing when non-iconic images are used, or (e) that we simply do not understand what tests of unfamiliar face processing are actually tapping. Thus, the findings reported here could be interpreted as evidence that famous face tests should carry relatively more weight in diagnosis, at least until we have a more fluent understanding of other diagnostic tools. 

In the meantime, given general agreement that diagnosis should follow evidence of impairment on at least two objective measures [4,12], the current findings suggest that some modification of existing protocols may be necessary. While face perception impairments may be more common than previously envisaged, they may not be severe enough to reach diagnostic cut-offs. Thus, at least two measures of face familiarity should be administered at the initial screening phase. However, given evidence of more highly prevalent selective impairments in famous face recognition, both short- and long-term face memory needs to be independently assessed, preferably each by two independent measures. Our finding that 17 control participants performed in the impaired range on any one test, and only one older individual performed below cut-off on multiple tests, supports this protocol. In terms of the testing battery itself, for now, an efficient solution would be to use the CFMT, CFMT-Aus [37] and famous face tests at screening. The CFMT-Aus provides a second measure of unfamiliar face recognition (analogous to the standard CFMT), as well as a delayed recall measure that could be compared to famous face performance. Face perception may then be later assessed in those who meet criteria, if theoretically and/or practically relevant. Importantly, famous face tests could be modified from the format used here (and in most papers within the field), to include measures of face perception, familiarity, semantics, and naming. 

However, one subgroup of individuals where famous faces tests are impractical for the diagnosis of DP is children, who are less likely to be exposed to a homogenous group of famous faces. Consequently, there is a pressing need to adapt existing diagnostic batteries for children to incorporate tests of long-term face memory (potentially through adapting current CFMT-based formats to include delayed recall measures, as in the CFMT-Aus). Incorporating these measures has significant practical benefits, as it would ensure that children and adults with DP are comparable in terms of their face recognition deficits, and facilitate cross-sectional and longitudinal studies in the area. Furthermore, effective assessment of long-term memory in children with face recognition impairments would allow researchers to develop a better theoretical understanding of the developmental trajectory of different aspects of face processing and identification.

Importantly, our findings underline the importance of objective testing in DP diagnosis. Even if we are very liberal and accept that the additional 42 individuals who were independently impaired at famous face recognition may represent an additional subtype of DP, approximately a third of the sample still failed to reach any inclusion criteria. It is likely that these people are either misjudging their face recognition ability, or that they experience less severe face recognition difficulties that are not captured by binary approaches to diagnosis. Indeed, it is possible that DP resides on the end of a single face recognition continuum, and this concept is also supported by the varied performance observed in our controls (see Figure 1 and Figure 2). While further work needs to explore this issue and map the severity of real-world functional experiences onto diagnostic test scores, it is likely that a reliable mapping cannot currently be achieved. This may be because (a) existing assessment tasks are not capturing a critical aspect of real-world face recognition performance, and/or (b) some individuals (both those with and without impairment) have little insight into their face recognition skills. Until this issue is resolved, and diagnostic tasks are developed that prohibit compensatory strategies, it would be prudent for diagnostic approaches to heed both objective and subjective information (for a recent discussion, see [62]). This may particularly be necessary for older adults, where self-report may need to carry more weight in diagnosis, alongside famous face and/or bespoke personally familiar face recognition tests. Indeed, the progressively lower means and larger standard deviations in older adults make it very difficult to infer impairment on the CFMT and CFPT (akin to the findings of Bowles et al.: [2]), and this may have prohibited diagnosis of some of the older adults in our sample.

## 5. Conclusions

In sum, the pattern of findings reported here suggests that current diagnostic protocols are missing a substantial number of impaired cases. Whether this results from methodological issues within existing tests, a lack of tests that tap different aspects of face-processing, or limitations in our current understanding of different subtypes of DP has yet to be clarified. We therefore recommend that the field continues to develop new, reliable tests of face perception, and multiple tests of unfamiliar and familiar face memory that strive to unpick the complex process of face learning.

## Figures and Tables

**Figure 1 brainsci-09-00133-f001:**
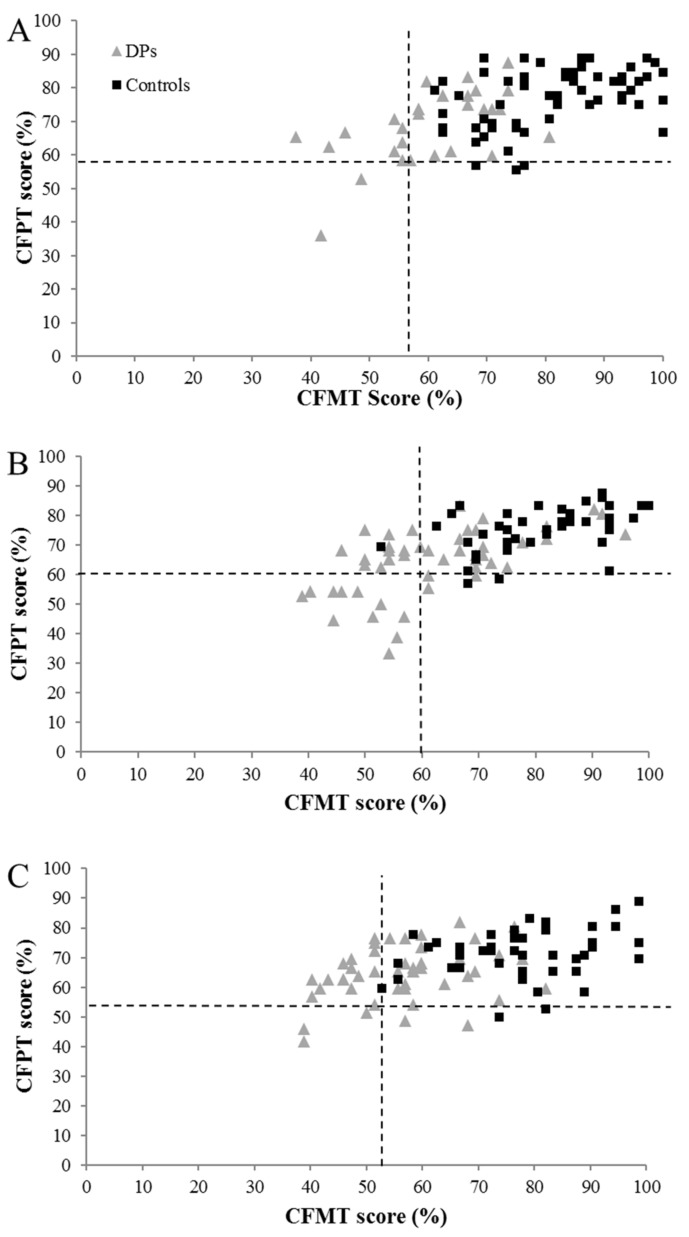
Relationship between CFMT and CFPT scores in the self-referred DP and control groups, for (**A**) 18–34 year olds, (**B**) 35–49 year olds, and (**C**) 50–59 year olds (those aged over 60 years are not displayed as more varied performance prohibits interpretation). Dotted lines represent two SDs from the control mean.

**Figure 2 brainsci-09-00133-f002:**
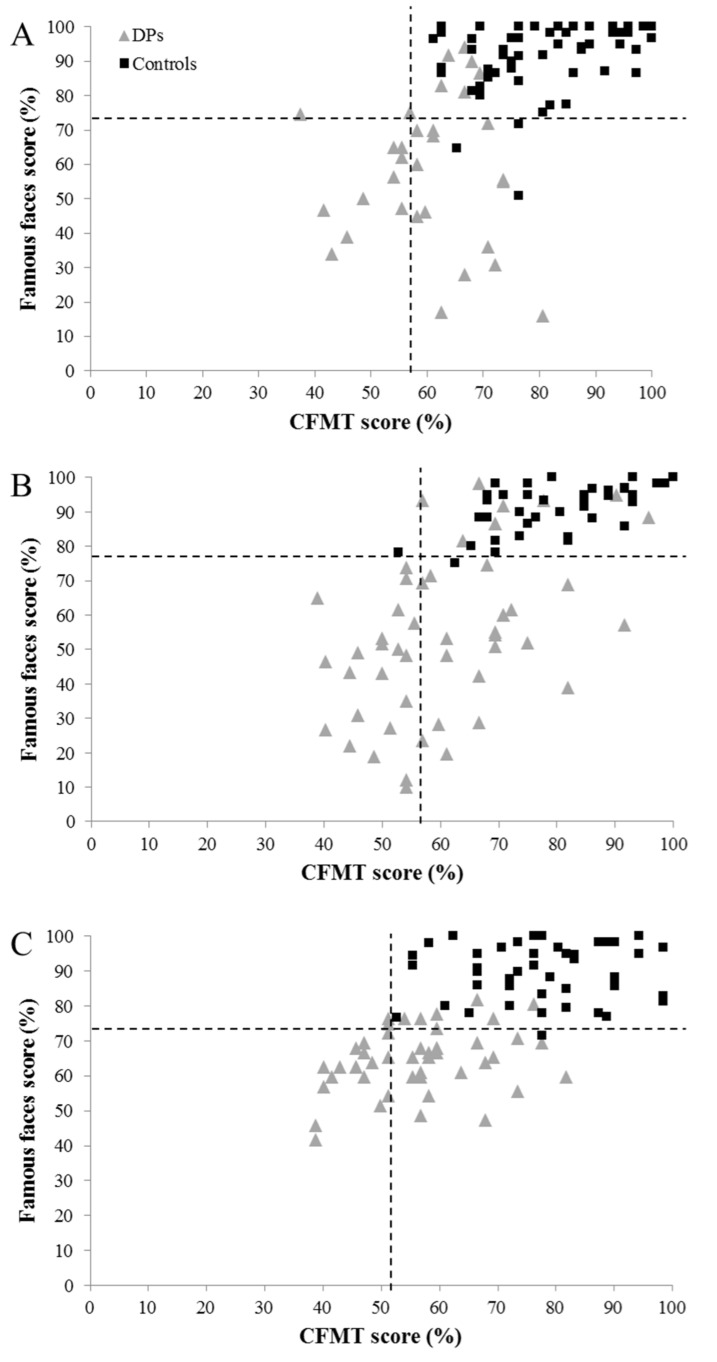
Relationship between CFMT and famous face scores in the self-referred DP and control groups, for (**A**) 18–34 year olds, (**B**) 35–49 year olds, and (**C**) 50–59 year olds (those aged over 60 years are not displayed as more varied performance prohibits interpretation). Dotted lines represent two SDs from the control mean.

**Figure 3 brainsci-09-00133-f003:**
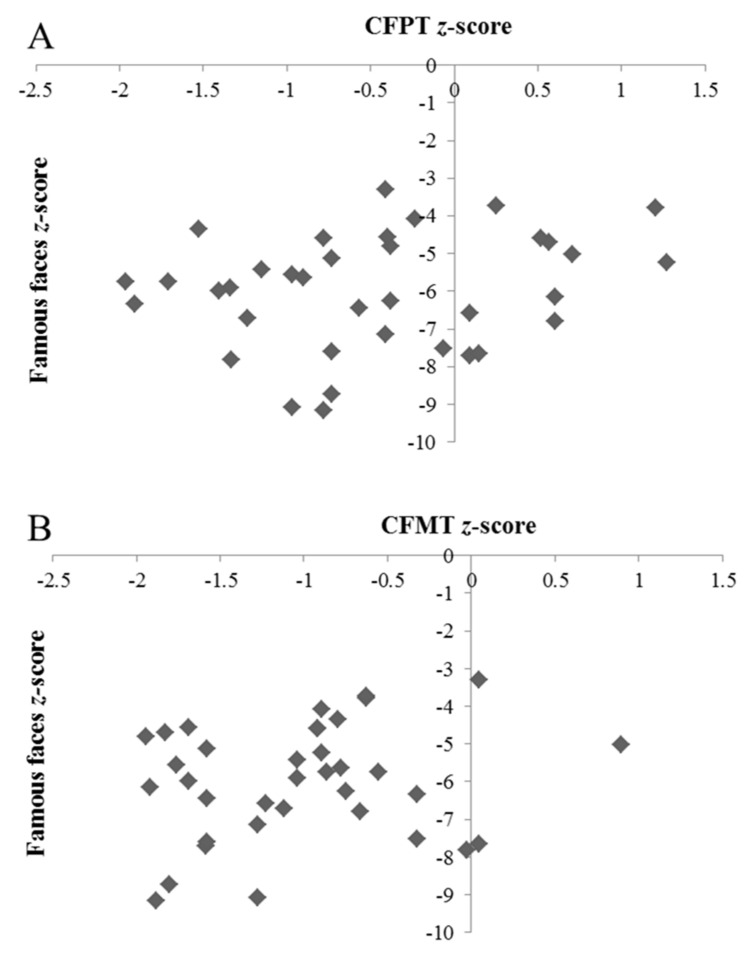
Relationship between famous face *z*-scores and (**A**) CFPT and (**B**) CFMT *z*-scores in DPs who showed significant dissociations between impaired famous face recognition and both intact CFMT and CFPT performance.

**Table 1 brainsci-09-00133-t001:** Demographic information for control participants from each age group, including mean (SD) age and IQ (measured using the Wechsler test of adult reading (WTAR)). Mean (SD) percentage accuracies for the Cambridge face perception test (CFPT), Cambridge face memory test (CFMT) and famous face test (FFT) are also presented.

Age Group (Years)	*N* (% Female)	Age	IQ	CFPT	CFMT	FFT
18–34	62 (53.2)	24.3 (4.2)	113.9 (8.9)	76.95 (8.77)	80.85 (11.57)	91.92 (9.74)
35–49	44 (56.8)	42.0 (4.5)	111.6 (8.8)	75.28 (7.47)	81.38 (11.52)	92.04 (6.96)
50–59	43 (65.1)	54.5 (2.9)	110.7 (9.5)	71.38 (8.30)	77.52 (12.17)	89.54 (8.04)
60–69	52 (61.5)	64.4 (2.8)	117.3 (8.3)	70.33 (7.64)	73.13 (13.56)	88.73 (9.58)
70–79	40 (57.5)	73.5 (2.3)	118.1 (7.2)	63.33 (10.07)	63.37 (12.51)	75.21 (13.14)

**Table 2 brainsci-09-00133-t002:** Control patterns of impaired performance on the CFPT, CFMT and famous faces test (FFT).

Age Group	*N*	All	CFPT and CFMT	CFPT and FFT	CFMT and FFT	CFPT Only	CFMT Only	FFT Only	None
18–34	62	0	0	0	0	3	0	3	56
35–49	44	0	0	0	0	2	1	1	40
50–59	43	0	0	0	0	2	0	1	40
60–69	52	0	0	0	0	2	0	1	49
70–79	40	0	1	0	0	1	0	0	38
Total	241	0	1	0	0	10	1	6	223

**Table 3 brainsci-09-00133-t003:** Sample size of developmental prosopagnosia (DPs) in each age group, and number of participants that were impaired on the CFPT, CFMT and famous faces test (FFT).

Age Group	*N* (% Female)	All	CFPT and CFMT	CFPT and FFT	CFMT and FFT	CFPT Only	CFMT Only	FFT Only	None
18–34	31 (64.5)	3	1	0	6	0	1	14	6
35–49	51 (72.5)	11	1	3	13	0	0	15	8
50–59	48 (68.8)	4	0	2	11	1	4	21	5
60–69	27 (63.0)	2	0	5	1	0	0	10	9
70–79	8 (50.0)	0	0	0	0	0	0	3	5
Total	165	20	2	10	31	1	5	63	33

**Table 4 brainsci-09-00133-t004:** Varimax rotated component loadings for the entire self-referred DP group’s (*N* = 165) performance on the CFPT, CFMT and famous face test (FFT).

Component	1	2
CFPT	0.84	
CFMT	0.86	
FFT		0.98

**Table 5 brainsci-09-00133-t005:** Pearson’s correlations for the entire self-referred DP group’s (*N* = 165) performance on the CFPT, CFMT and famous face test (FFT).

	CFPT	CFMT	FFT
CFPT	1	0.51 *	0.35 *
CFMT		1	0.35 *
FFT			1

*, *p* < 0.001 (sequential Bonferroni correction applied).

**Table 6 brainsci-09-00133-t006:** SDT results for cases with significant dissociations between intact CFPT performance and impaired CFMT and famous face test (FFT) scores. The sequential Bonferroni correction is applied to correct for multiple comparisons.

Case	% Correct			CFPT v CFMT			CFPT v FFT		
	CFPT	CFMT	FFT	*t*	*p*	% More Extreme	*t*	*p*	% More Extreme
DP032	68.06	45.83	30.91	2.04	0.05	2.37	5.77	0.001	0.01
DP036	68.06	45.83	30.91	2.04	0.05	2.37	5.77	0.001	0.01
DP042	73.61	54.17	10.00	2.06	0.05	2.27	8.50	0.001	0.01
DP054	75.00	50.00	43.10	2.59	0.01	0.66	5.16	0.001	0.01

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
