# Peer review of "Objective Patterns of Face Recognition Deficits in 165 Adults with Self-Reported Developmental Prosopagnosia"

_brainsci, 2019, doi:10.3390/brainsci9060133_

Reviewer 1 Report

This is an informative paper with important implications surrounding the diagnosticity of Developmental Prosopagnosia. Interestingly, the findings suggest that DP, like general face recognition ability, may be characterized by a range of individual differences, whereby some DPs are poor at some face tasks but not others.

The paper is well-written and the results are effectively communicated. I have no concerns that should preclude its publication in Brain Sciences. 

Author Response

Thank you for your positive feedback on our manuscript.

No further comments are required as no changes required by this reviewer.

Reviewer 2 Report

The authors study the patterns of covariance and dissociation of face perception, short-term unfamiliar face recognition, and long-term familiar face recognition abilities in a large (n=165) sample of persons self-identifying as having severe everyday difficulties with face recognition (i.e., suspected developmental prosopagnosia).  The authors compare these to control participants’ abilities (n = 241).  The key finding is that a quarter of those in the sample with putative developmental prosopagnosia had impaired famous (i.e., long-term familiar) face recognition that dissociated from their intact short-term unfamiliar face recognition and face perception abilities, as defined by standard cut-off scores on the best available standardized tests for prosopagnosia and by statistical comparisons with the control sample’s scores.  The authors propose that this finding is consistent with the idea that there may be a sub-type of developmental prosopagnosia in which long-term familiar facial recognition is impaired, but short-term unfamiliar face recognition and face perception are unimpaired (or at least not severely impaired).  Importantly, the implication of this possibility is that a large proportion of such persons with developmental prosopagnosia would not be diagnosed appropriately (i.e., would receive a false negative), since most researchers define developmental prosopagnosia as requiring evidence of impairment on at least two of the three standardized tests used in the present study.  The authors do, however, also present considerations and potential limitations of these commonly used standardized tests that could also account for their finding.  The authors provide a helpful discussion on these issues that is much needed and valuable in it’s own right.

I would like to commend the authors for a very nice study and a well written paper.  I find the evidence presented in the paper suggests that the proposed hypothesis of a previously unrecognized common sub-type of developmental prosopagnosia is a promising hypothesis.  Given its potential implications for the diagnosis and the mechanistic understanding of prosopagnosia, the hypothesis is worthy of future verification and investigation, and could constitute a significant conceptual advance for the field.  Even if this hypothesis turns out to be false upon further investigation, that line of inquiry would likely at least elucidate how the diagnostic assessments and criteria for developmental prosopagnosia could be improved.

I believe the paper would be more than acceptable to publish in its current form.  I would, however, like to take the opportunity to make a couple minor remarks for merely the authors’ consideration.

1) A large proportion of those reporting face recognition difficulties did not meet criteria for impairment on any of the tests.  How do you interpret this finding?  Do some of those individuals have a weaker form of prosopagnosia and others have something like an “imposter syndrome” for their face recognition abilities?  Much of the paper at least implicitly assumes some some kind of binary threshold for developmental prosopagnosia as an ideal (e.g., use of cutoff scores, establishment of dissociations on the basis of statistical thresholds, discussion of potentially adjusting cutoff scores for either CFMT/CFPT or FFT, discussion of whether impairment of one domain vs. two should suffice, etc.).  I note though that in Figures 1 and 2 there is a fair amount of overlap between the “DPs” and the controls indicating a continuum of facial recognition abilities.  Beyond the diagnostic and definitional issues of developmental prosopagnosia already raised in the paper, do you think your results may also have implications for whether we think of developmental prosopagnosia as a binary vs. spectrum phenomenon (or a spectrum with dissociable dimensions)?

2). Very relatedly, among the diagnostic and definitional issues of prosopagnosia raised in the present paper, one that did not get much treatment was the issue of insight into one’s condition.  The sample of “DPs” of the current study (and in most DP studies) would be persons who have high insight into their condition because of how they were (self)selected.  Also, a few control participants showed evidence of impairment on at least one assessment.  Though it is reasonable to consider those in the controls as occurring “by chance” because of the relatively small number of those cases, it is not impossible that a few of the control participants had developmental prosopagnosia, but had low/no insight into their condition.  I realize that a subjective assessment of the severity of ones own facial recognition deficit was not included in the battery of tests of this study, so it is not possible to investigate such questions of insight (or “imposter syndrome”) with your data.  However, as speculation, what value, if any, do you think such self-reported assessment may have in the diagnosis and understanding of developmental prosopagnosia?  Incidentally, I recently published a paper attempting to tackle this issue of insight, in addition to other issues similar to those raised in the present paper, that you may find interesting (Arizpe, et al, 2019, Behavioral Research Methods, https://doi.org/10.3758/s13428-018-01195-w).  We have also made our data from that study publicly available so anyone interested in these issues can access the data themselves (n = 1,518 from general population, CMFT, FFT, +subjective assessment).

Author Response

Thank you for your positive and constructive review of our paper. Our replies to the two suggestions are copied below:

1) A large proportion of those reporting face recognition difficulties did not meet criteria for impairment on any of the tests.  How do you interpret this finding?  Do some of those individuals have a weaker form of prosopagnosia and others have something like an “imposter syndrome” for their face recognition abilities?  Much of the paper at least implicitly assumes some some kind of binary threshold for developmental prosopagnosia as an ideal (e.g., use of cutoff scores, establishment of dissociations on the basis of statistical thresholds, discussion of potentially adjusting cutoff scores for either CFMT/CFPT or FFT, discussion of whether impairment of one domain vs. two should suffice, etc.).  I note though that in Figures 1 and 2 there is a fair amount of overlap between the “DPs” and the controls indicating a continuum of facial recognition abilities.  Beyond the diagnostic and definitional issues of developmental prosopagnosia already raised in the paper, do you think your results may also have implications for whether we think of developmental prosopagnosia as a binary vs. spectrum phenomenon (or a spectrum with dissociable dimensions)?

Thank you for raising these issues. The issue of whether DP is a categorical or dimensional condition is theoretically important and has not yet been resolved. Unfortunately the data reported here cannot answer this question in a robust way, but we acknowledge that many researchers take the "continuum" perspective, although there has been no substantial attempt to date to grade real-world severity with scores on objective tests. This most likely results from an absence of reliable tools that can either quantify subjective experience or objective performance - particularly given the forced-choice nature of tests such as the CFMT. Our paper aimed to focus on objective tests, and, as the reviewer acknowledges below, we didn't collect any graded scores on real-world severity. On the basis of the available data and the limitations of existing tools we do not feel we can speculate on dimensionality in a meaningful way - likely rather a different information-processing approach is needed to assess the qualitative processing of faces. Nevertheless, we have amended the manuscript to acknowledge this important question (p. 14, para 4) and thank the reviewer for the thoughtful comment.

2). Very relatedly, among the diagnostic and definitional issues of prosopagnosia raised in the present paper, one that did not get much treatment was the issue of insight into one’s condition.  The sample of “DPs” of the current study (and in most DP studies) would be persons who have high insight into their condition because of how they were (self)selected.  Also, a few control participants showed evidence of impairment on at least one assessment.  Though it is reasonable to consider those in the controls as occurring “by chance” because of the relatively small number of those cases, it is not impossible that a few of the control participants had developmental prosopagnosia, but had low/no insight into their condition.  I realize that a subjective assessment of the severity of ones own facial recognition deficit was not included in the battery of tests of this study, so it is not possible to investigate such questions of insight (or “imposter syndrome”) with your data.  However, as speculation, what value, if any, do you think such self-reported assessment may have in the diagnosis and understanding of developmental prosopagnosia?  Incidentally, I recently published a paper attempting to tackle this issue of insight, in addition to other issues similar to those raised in the present paper, that you may find interesting (Arizpe, et al, 2019, Behavioral Research Methods, https://doi.org/10.3758/s13428-018-01195-w).  We have also made our data from that study publicly available so anyone interested in these issues can access the data themselves (n = 1,518 from general population, CMFT, FFT, +subjective assessment).

Again, we would like to thank the reviewer for this thoughtful comment. We did address the issue of self-report to some extent in the initial draft of the manuscript (p. 14, para 4), and we have now extended this discussion in the same paragraph, and included the recommended citation. Again, as we state in our response to Point 1, we do not feel that we can speculate too strongly on this issue based on our data - which perhaps raises more questions about diagnosis than it answers. Indeed, it is unclear whether such a large proportion of people did not reach objective diagnostic criteria for DP because (a) they have a lack of insight, (b) the diagnostic criteria are incorrect and do not allow for self-report, or (c) the currently available objective tests are not accurate enough/allow compensatory processing/do not capture real-world face recognition abilities. We have now included this in our discussion (p. 14, para 4), and re-highlighted our existing recommendation that self-report may be particularly necessary for the diagnosis of older adults, for whom computerized tests are less reliable. We hope this addresses the reivewer's thoughts as much as we feel able to from the current data.

Many thanks once again for these useful and insightful comments.